

# Imbalance between subsets of CD8+ peripheral blood T cells in patients with chronic obstructive pulmonary disease

Long Chen[1,*], Gang Chen[1,*], Ming-Qiang Zhang[1,2], Xian-Zhi Xiong[1], Hong-Ju Liu[1], Jian-Bao Xin[1], Jian-Chu Zhang[1], Jiang-Hua Wu[1], Zhao-Ji Meng[1] and Sheng-Wen Sun[1]

[1] Department of Respiratory Medicine, Union Hospital, Tongji Medical College, Huazhong University of Science and Technology, Wuhan, China
[2] Department of Respiratory Medicine, Beijing Tsinghua Changgung Hospital, Medical Center of Tsinghua University, Beijing, China
[*] These authors contributed equally to this work.

## ABSTRACT

**Background.** CD8+ T lymphocytes are known to play a critical role in the pathogenesis of chronic obstructive pulmonary disease (COPD). However, systematic analyses of CD8+ T cell (Cytotoxic T cells, Tc) subsets in COPD patients have yet to be well conducted.

**Methods.** The whole Tc subsets, including Tc1/2/10/17, CD8+ regulatory T cells (Tregs) and CD8+$\alpha$7+ T cells, were quantified by flow cytometry in peripheral blood from 24 stable COPD subjects (SCOPD), 14 patients during acute exacerbations (AECOPD), and 14 healthy nonsmokers (HN).

**Results.** Acute exacerbations of COPD were accompanied by elevated levels of circulating CD8+ T cells. Tc1 cells were increased in both SCOPD and AECOPD patients, whereas the percentage of Tc2 cells was decreased in SCOPD patients but remained normal in AECOPD patients. Tc17 cells were increased only in AECOPD patients, and the percentage of Tc10 cells was reduced in both SCOPD and AECOPD patients. The imbalances of pro/anti-inflammatory Tc subsets observed in COPD may be caused by the lack of Tc10 cells and the impaired anti-inflammatory capacity of CD8+ Tregs.

**Conclusions.** The imbalances between subsets of CD8+ peripheral blood T cells contribute to the immune response dysfunction in COPD pathogenesis.

## INTRODUCTION

Chronic obstructive pulmonary disease (COPD) is characterized by poorly reversible airflow limitation and progressive airway inflammation. Currently, no cure or effective treatment is available to stop the progression of COPD. Airway inflammation is triggered by the inhalation of a variety of hazardous gases and particles, with tobacco smoke serving as the leading contributing factor (*Hogg et al., 2004*; *Lakshmi et al., 2014*; *Mortaz et al., 2008*).

Corresponding author
Xian-Zhi Xiong,
xxz0508@hust.edu.cn

Increasing evidence indicates that T cell-mediated inflammation, as a key component, is involved in the pathogenesis of COPD (*Saetta et al., 1993*). Previous studies have shown that the numbers of CD8[+] T cells (Cytotoxic T cells, Tc) in the lung parenchyma and small airways are inversely correlated with pulmonary function (*O'Shaughnessy et al., 1997*; *Saetta et al., 1999*; *Saetta et al., 1998*), and depleting these cells may protect against alveolar destruction due to chronic cigarette smoke exposure (*Podolin et al., 2013*); together, these findings suggest that CD8[+] T lymphocytes may play a critical role in COPD pathogenesis. Several studies have focused on CD8[+] T cells in the blood of COPD patients (*Freeman et al., 2015*; *Makris et al., 2008*; *Mathai & Bhat, 2013*); however, a comprehensive study on circulating CD8[+] T cell subsets in patients with this disease has yet to be conducted. Although some *in vitro* research has indicated that soluble components extracted from cigarette smoke can significantly reduce T cell activation and proliferation (*Glader et al., 2006*; *Lambert et al., 2005*), substantial evidence has shown that large numbers of CD8[+] T cells are present in the airways and parenchyma of smokers with COPD (*Saetta et al., 1999*). Thus, the precise influence of cigarette smoke on CD8[+] T cells remains unclear.

Although pro-inflammatory CD8[+] T cell pool (mainly including CD8[+] IFN-$\gamma^+$Tc1, CD8[+] IL-4[+]Tc2, and CD8[+] IL-17[+]Tc17) and anti-inflammatory CD8[+] T cell pool (mainly including CD8[+] Foxp3[+]Treg, and CD8[+] IL-10[+]Tc10) have been considered to be involved in COPD pathogenesis, existing studies regarding this association vary widely in their designs, population sizes and experimental methods. Meanwhile, the $\alpha$7 nicotinic acetylcholine receptors ($\alpha$7 nAChRs) are recently discovered to be required for inhibition of TNF-$\alpha$ (*Wang et al., 2003*). However, the expression and function of $\alpha$7 nAChRs in CD8[+] T cell, especially in the development of COPD is yet to be illuminated.

In the present study, we attempted to investigate systemic CD8[+] T cell subsets, including Tc1/2/10/17, CD8[+] regulatory T cells (Tregs) and CD8[+]$\alpha$7[+] T cells, from healthy nonsmokers and patients with either stable COPD (SCOPD) or during acute exacerbations (AECOPD). Especially, the data shown here are derived from the same sample set published earlier dealing with an in-depth analysis of the CD4[+] T cell compartment in COPD patients (*Zhang et al., 2014*). To the best of our knowledge, this is the first report showing the complete subsets of circulating CD8[+] T cells in the pathogenesis of COPD.

## MATERIALS AND METHODS

### Subjects

This study was approved by the Ethics Committee of Tongji Medical School, Huazhong University of Science and Technology (# 2013/S048), and was conducted in accordance with the Declaration of Helsinki. Informed written consent was obtained from all subjects. The demographic characteristics of subjects clinical characteristics have been presented in detail previously (*Zhang et al., 2014*). In brief, according to the diagnostic criteria from the Global Initiative for Chronic Obstructive Lung Disease (GOLD) (*Rabe et al., 2007*), we collected 24 SCOPD (mean age, 66.5 $\pm$ 7.2 years; smoking history, 41.9 $\pm$ 17.6 pack-years) and 14 AECOPD patients (mean age, 69.1 $\pm$ 9.6 years; smoking history, 47.2 $\pm$ 19.5 pack-years). Meanwhile, 14 healthy nonsmokers (HN; mean age, 65.6 $\pm$ 7.8 years) without any lung

or systemic disease were also recruited. Patients with SCOPD were free of exacerbation for at least 4 weeks at the time of blood draw. AECOPD patients were identified by the initiation of symptoms diagnostic for COPD exacerbation in the past three days without any new therapeutic intervention. Exclusion criteria for the study included the following characteristics: other respiratory diseases apart from COPD, systemic autoimmune diseases and malignant tumors.

## Sample collection

Peripheral blood samples were collected in heparin-treated tubes from each subject within 24 h after arrival at the hospital. Blood samples were immediately placed on ice and then centrifuged at 1,200 g for 5 min. Then peripheral blood mononuclear cells (PBMCs) were isolated from the heparinized blood by Ficoll-Hypaque gradient centrifugation (Pharmacia, Uppsala, Sweden) as previously described (*Zhang et al., 2014*). Isolated peripheral blood cells were used for subsequent experiments.

## Flow cytometric analysis

The expression of surface and intracellular markers by T cells was assessed using flow cytometry as previously described (*Zhang et al., 2014*). Cell staining was performed using the following anti-human-specific antibodies (Abs): anti-CD3 PerCP-Cy5.5 (clone OKT3; eBioscience, San Diego, CA, USA), anti-CD8 FITC (Clone RPA-T8; BD Biosciences, San Jose, CA, USA), anti-Foxp3 PE (Clone 236A/E7; eBioscience), anti–IFN-$\gamma$ PE (Clone 4S.B3; eBioscience), anti-IL-4 PE-Cy7 (Clone 8D4-8; eBioscience), anti-IL-17A eFluor660 (Clone eBio64DEC17; eBioscience), and anti-IL-10 Alexa Fluor647 (Clone JES3-9D7; eBioscience). Meanwhile, the expression of $\alpha$7 AChRs on T cells was detected by its binding to $\alpha$-bungarotoxin labeled with Alexa Fluor647 (Invitrogen, Carlsbad, CA, USA). Intracellular cytokines staining was performed after the T cells were stimulated with PMA (50 ng/ml; Sigma-Aldrich, St. Louis, MO, USA) and ionomycin (1 $\mu$M; Sigma-Aldrich) in the presence of GolgiStop (BD Biosciences) for 5 h. Appropriate species matched Abs were used as isotype controls. Lymphocyte population was discriminated in the light-scattering measurement (FSC/SSC dot plot) by surrounding them with a polygon gate to exclude debris and dead/adhesive cells. Flow cytometry was performed on a FACS Canto II (BD Biosciences) and analyzed using BD FCSDiva Software and FCS Express 4 software (De Novo Software, Los Angeles, CA, USA).

## Statistical analysis

Data are expressed as the mean $\pm$ SD (unless indicated in the figure legends). Comparisons of data between different groups were performed using a Kruskal-Wallis one-way analysis of variance (ANOVA) based on rank, with Tukey and Dunn's post-hoc tests for between-group comparisons. Data analysis was performed using GraphPad Prism v.5.01 software (GraphPad Software, La Jolla, CA, USA), and two-tailed $P$ values of less than 0.05 were considered statistically significant.

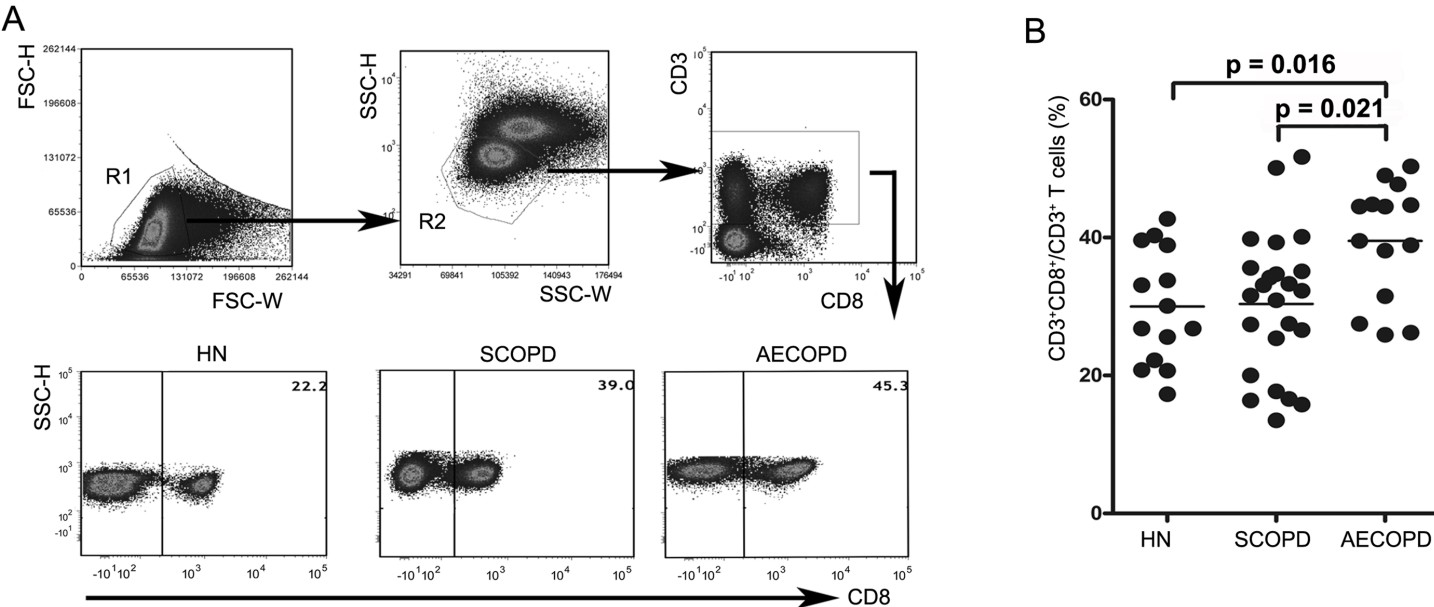

**Figure 1** **Acute exacerbations of COPD are accompanied by elevation of circulating CD8$^+$ T cells.** (A) Strategies for gating CD8$^+$ T cells: PBMCs were gated into R1 and R2 gates to exclude monocytes, platelets and debris; CD8$^+$ T cells were then identified based on CD3 and CD8 expression. Representative flow cytometric dot plots are shown. (B) Comparisons of the percentages of CD8$^+$ T cells from HN , SCOPD patients and AECOPD patients ($n = 14$, 24, and 14, respectively) are shown. Horizontal bars indicate the mean values.

## RESULTS

### Acute exacerbations of COPD are accompanied by elevated levels of circulating CD8$^+$ T cells

We first determined the percentages of CD8$^+$ T cells in blood samples obtained from healthy nonsmokers and SCOPD and AECOPD patients using flow cytometry (Figs. 1A and 1B). We found that AECOPD patients ($39.51 \pm 8.55\%$) showed the highest percentage of CD8$^+$ T cells in the peripheral blood compared with the other two groups, with both comparisons reaching statistical significance. However, the percentage of CD8$^+$ T cells in SCOPD patients ($30.36 \pm 10.25\%$) was similar to that in healthy nonsmokers ($29.99 \pm 8.49\%$).

### Imbalance of peripheral Tc1, Tc2 and Tc17 cells in COPD

To investigate changes in CD8$^+$ T cell subsets in COPD patients, we analyzed the following subsets of T cells in the blood: CD3$^+$CD8$^+$IFN-$\gamma^+$ (Tc1), CD3$^+$CD8$^+$IL-4$^+$ (Tc2), CD3$^+$CD8$^+$IL-17A$^+$ (Tc17), CD3$^+$CD8$^+$Foxp3$^+$ (CD8$^+$ Tregs), CD3$^+$CD8$^+$IL-10$^+$ (Tc10) and CD3$^+$CD8$^+\alpha7^+$ (CD8$^+\alpha7^+$) (Figs. 2 and 3).

As shown in Figs. 2A and 2B, Tc1 cells were markedly elevated in both the SCOPD ($37.10 \pm 21.26\%$) and AECOPD ($53.11 \pm 18.03\%$) patients compared with the healthy nonsmokers ($17.56 \pm 13.90\%$). Furthermore, the frequency of Tc1 cells was higher in AECOPD patients than in SCOPD patients. Similar to Tc1 cells, the percentage of Tc17 cells was highest in AECOPD patients ($1.13 \pm 0.83\%$); however, the levels of Tc17 cells were similar between healthy nonsmokers and SCOPD patients ($0.33 \pm 0.12\%$ and $0.38 \pm 0.38\%$, respectively). Additionally, the percentage of Tc2 cells was significantly reduced in

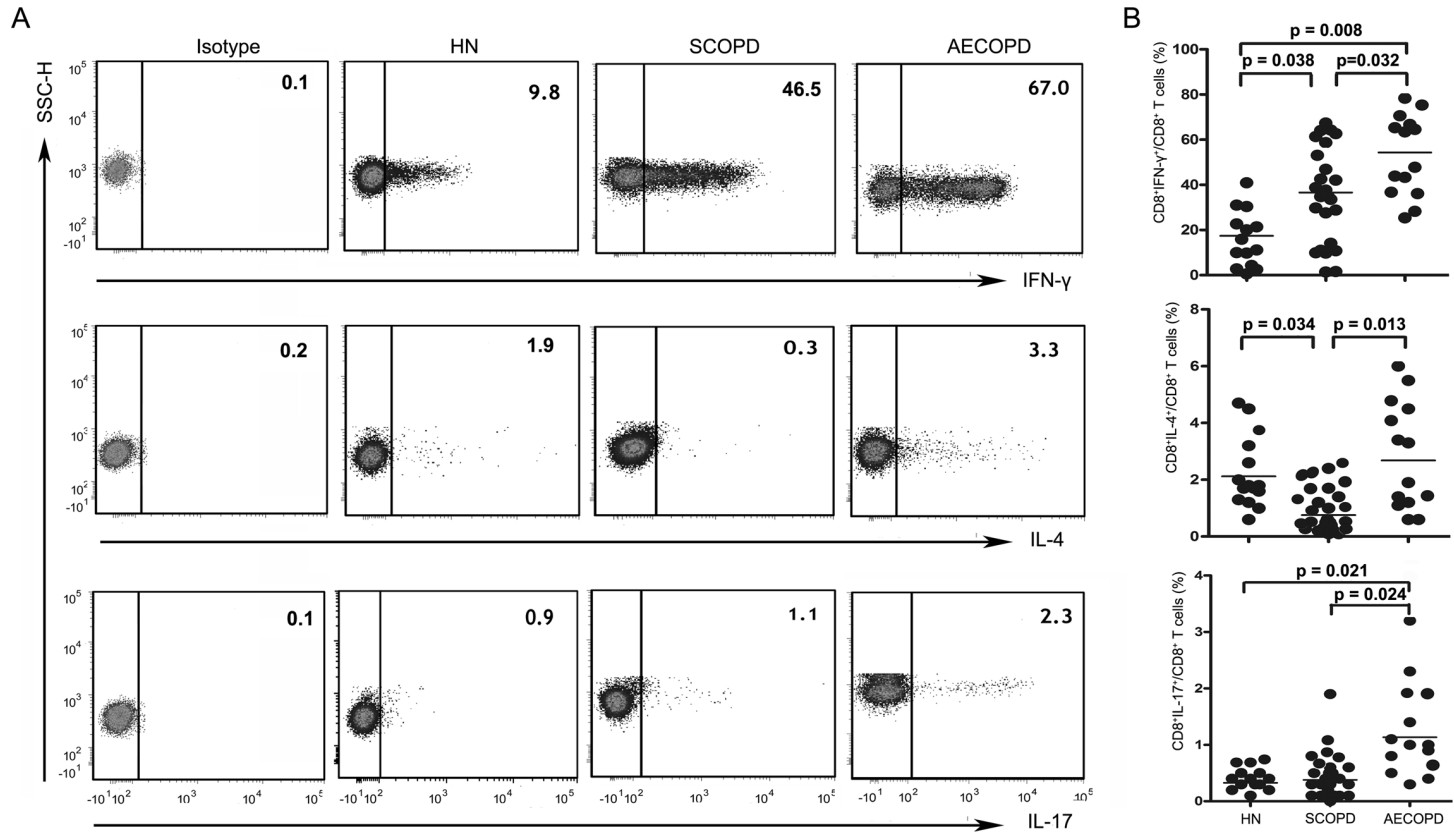

**Figure 2** **Imbalance of peripheral Tc1, Tc2 and Tc17 cells in COPD.** (A) Representative flow cytometric dot plots of Tc1, Tc2 and Tc17 cells (Tc1: CD3$^+$CD8$^+$IFN-$\gamma^+$ T cells, Tc2: CD3$^+$ CD8$^+$ IL-4$^+$ T cells, Tc17: CD3$^+$ CD8$^+$ IL-17A$^+$ T cells) in HN, SCOPD patients and AECOPD patients are shown. (B) Comparisons of the percentages of Tc1, Tc2 and Tc17 cells in HN, SCOPD patients and AECOPD patients ($n = 14$, 24, and 14, respectively) are shown. Horizontal bars indicate the mean values.

SCOPD patients (0.76 ± 0.69%) compared with healthy nonsmokers and AECOPD patients (2.12 ± 1.22% and 2.68 ± 1.96%, respectively).

## Decreased Tc10 percentage in both SCOPD and AECOPD patients

The levels of CD8$^+$ Tregs appeared to be increased in SCOPD and AECOPD patients (2.05 ± 1.93% and 2.56 ± 2.17%, respectively) compared with healthy nonsmokers (0.76 ± 0.48%), although the difference was not statistically significant. Meanwhile, the percentages of CD8$^+\alpha7^+$ T cells in SCOPD and AECOPD patients (0.85 ± 0.83% and 0.59 ± 0.38%, respectively) showed a trend toward up-regulation when compared with healthy controls (1.03 ± 0.91%), but this was not statistically significant.

Unsurprisingly, as can be seen in Fig. 3, the frequencies of Tc10 cells in SCOPD and AECOPD patients (0.45 ± 0.54% and 0.42 ± 0.21%, respectively) were significantly reduced to half the level observed in healthy nonsmokers (1.06 ± 0.34%).

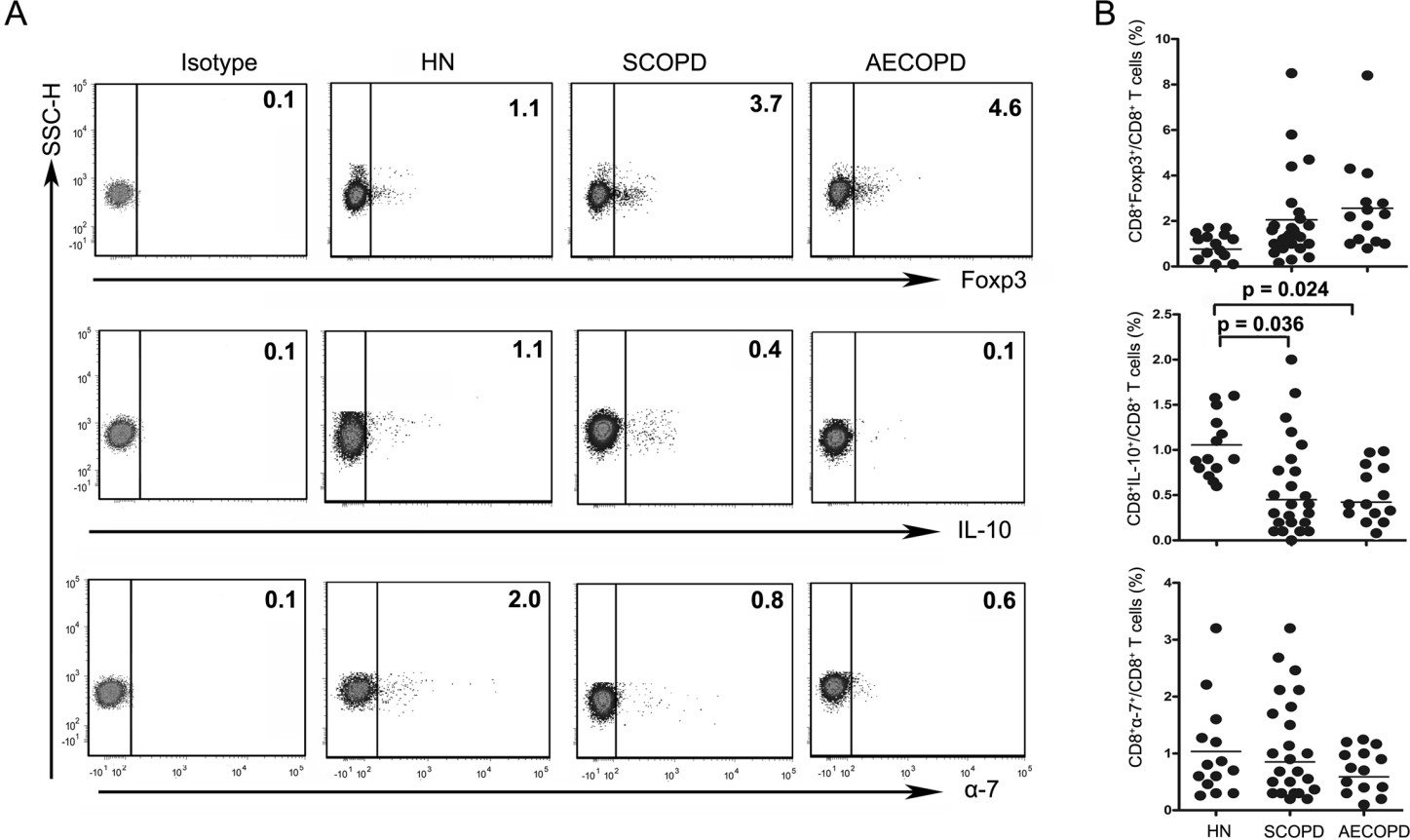

**Figure 3** **The percentage of Tc10 cells is decreased in both the SCOPD and AECOPD groups.** (A) Representative flow cytometric dot plots of CD8$^+$ Tregs, Tc10 and CD8$^+\alpha7^+$ T cells (CD8$^+$ Tregs: CD3$^+$CD8$^+$FOXP3$^+$ T cells; Tc10: CD3$^+$CD8$^+$IL-10$^+$ T cells; CD8$^+\alpha7^+$: CD3$^+$CD8$^+\alpha7^+$ T cells) in HN, SCOPD patients and AECOPD patients are shown. (B) Comparisons of the percentages of CD8$^+$ Tregs, Tc10 cells, and CD8$^+\alpha7^+$ T cells in HN, SCOPD patients and AECOPD patients ($n = $ 14, 24, and 14, respectively) are shown. Horizontal bars indicate the mean values.

## Comprehensive analysis of the relative percentages of CD8$^+$ T cell subsets

To precisely clarify which changes occur in different CD8$^+$ T cell subsets in COPD, we performed a comprehensive analysis of the relative proportion from mean value of each subset. However, since the immunoregulatory function and key transcription factors of CD8$^+\alpha7^+$ T cells have not been elucidated, we could not firmly establishing CD8$^+\alpha7^+$ population as an independent cytotoxic T cell lineage in human. Moreover, CD8$^+\alpha7^+$ T cells could even overlap with other subsets, so this population is disregarded in the summary Fig. 4.

As shown in Fig. 4, Tc1 cells accounted for the majority of pro-inflammatory CD8$^+$ T cells in both SCOPD and AECOPD patients (91% and 89%, respectively), and both of these values were higher than that observed in healthy nonsmokers (80%). Tc2 cells were radically reduced in SCOPD (2%) patients while only modestly reduced in AECOPD (4%) patients compared with healthy nonsmokers (10%). Notably, Tc17 cells were not increased in SCOPD or AECOPD patients (1% and 2%, respectively) compared with the healthy

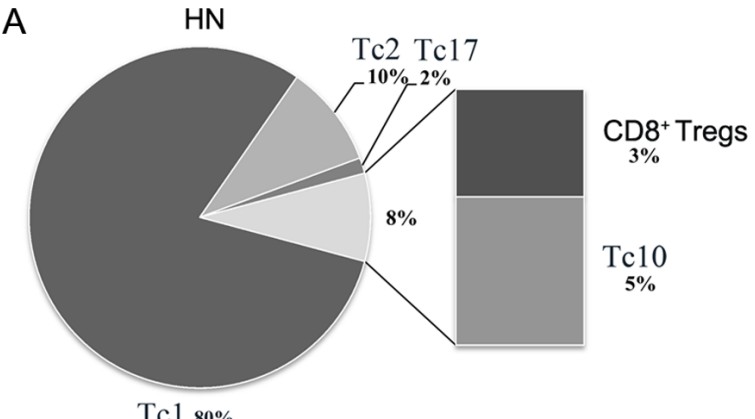

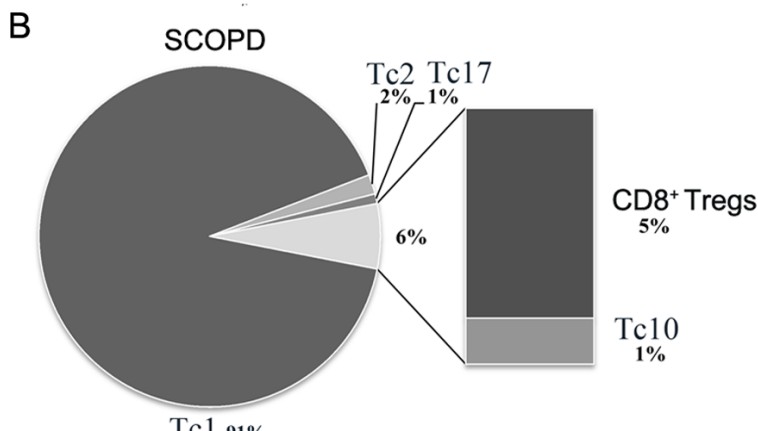

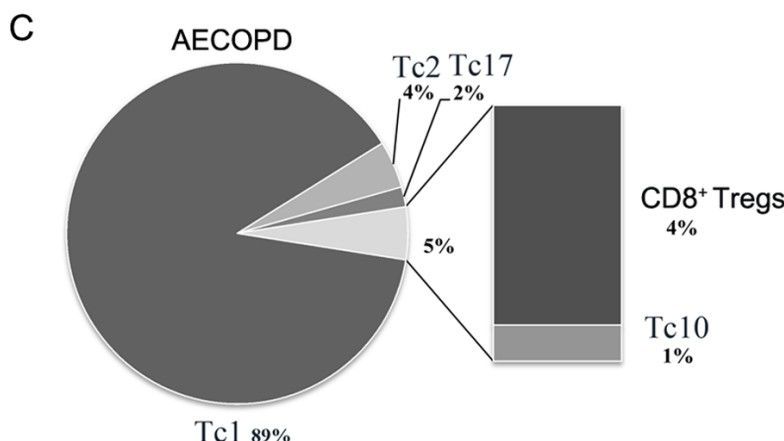

**Figure 4  Comprehensive analysis of the relative percentages of CD8+ T cell subsets.** The relative percentages of Tc1, Tc2, Tc17, CD8+ Tregs and Tc10 cells in HN (A), SCOPD patients (B) and AECOPD patients (C) were calculated based on their mean values. Anti-inflammatory CD8+ T cells comprised CD8+ Tregs and Tc10 cells. The data are presented in a pie graph.

nonsmokers (2%). We also observed that the percentage of anti-inflammatory CD8[+] T cells (CD8[+] Tregs and Tc10 cells) was highest in healthy nonsmokers (8%) but was reduced in SCOPD (6%) and progressively decreased in AECOPD patients (5%). Although CD8[+] Tregs were increased modestly in SCOPD (5%) and AECOPD (4%) patients compared with healthy nonsmokers (3%), Tc10 cells were remarkably decreased in both SCOPD (1%) and AECOPD (1%) patients compared with healthy nonsmokers (5%).

Together, this comprehensive analysis showed that, compared with healthy nonsmokers, relatively more Tc1 cells and CD8[+] Tregs, but less Tc2 and Tc10 cells, were found in both the SCOPD and AECOPD patients and that the overall population of anti-inflammatory CD8[+] T cells was reduced in both COPD groups.

## DISCUSSION

Previous studies have mainly focused on the sizes of the CD8[+] T cell pools and partial Tc subsets in the development of COPD (*Chang et al., 2011*; *Freeman et al., 2015*; *Makris et al., 2008*; *Saetta et al., 1999*). However, the collaborative relationship between overall Tc subsets is not well understood. To the best of our knowledge, this is the first study showing the complete subsets of circulating pro/anti-inflammatory CD8[+] T cells in the pathogenesis of COPD. Combined with our previous findings of an imbalance of pro/anti-inflammatory CD4[+] T pools in COPD patients, which revealed increased Th17 cells (only in AECOPD) and Th1 cells and reduced Th2 cells (only in SCOPD) and Th10 cells, as well as an impaired capacity of Tregs, our current research of CD8[+] compartment would contribute to the overall interpretation of the true role of T cell pools in COPD.

COPD is considered to be a multi-component disease with systemic manifestations in addition to local pulmonary inflammation. This inflammation contributes to repeated injury and repair cycles, finally resulting in the remodeling of tissue with altered structure and function (*Barnes, Shapiro & Pauwels, 2003*). In conflict with published findings (*Freeman et al., 2015*), the percentage of peripheral CD8[+] T cells was elevated in AECOPD but not in SCOPD patients. Combined with previous findings that the percentage of sputum CD8[+] T lymphocytes was significantly increased at the onset of exacerbations (*Makris et al., 2008*), we speculate that the increased frequency of CD8[+] T cells is limited to the lungs, whereas the expanded CD8[+] T lymphocyte population spreads throughout the body during an acute exacerbation.

Tc1 cells were also significantly elevated in the peripheral blood of both COPD groups, which was consistent with a previous report (*Zhu et al., 2009*). Interestingly, the percentage of circulating Tc2 cells was significantly decreased in SCOPD patients but returned to normal during an acute exacerbation. One prior study revealed that decreased Tc1/Tc2 cell ratios were found in the sputum at the onset of an exacerbation compared with the stable state (*Makris et al., 2008*). Another study however reported a greater Tc2 response in the bronchoalveolar lavage fluid of COPD patients (*Barcelo et al., 2006*). Different sample sources and experimental conditions could possibly account for this discrepancy. Tc17 cells have recently been identified as pro-inflammatory lymphocytes and are implicated in the pathogenesis of some human autoimmune diseases such as psoriasis (*Res et al., 2010*)
and systemic lupus erythematosus (*Henriques et al., 2010*). Tc17 cells are associated with cigarette smoke-induced lung inflammation and COPD (*Chang et al., 2011*; *Zhou et al., 2015*). Additionally, our results showed that the Tc17 cell population in the peripheral blood remarkably increased only in AECOPD patients; our findings indicate that Tc17 cells may participate in the acute exacerbation of COPD. From the perspective of the immune system, COPD mainly involves a Tc1 reaction; Tc2 and Tc17 reactions are attenuated in stable COPD. The above observations indicate that in the inhibitory state of the acute inflammatory reaction in COPD patients during the stable period, Tc2 and Tc17 cell levels remain unchanged unless the patient suffers from an acute irritation such as an infection.

We also assessed the anti-inflammatory $CD8^+$ T cells, including $CD8^+Foxp3^+Tregs$ and Tc10 in both COPD groups. $\alpha 7$ nAChR activation was recently found to be capable of suppressing inflammation by inhibiting specific cytokines such as TNF-$\alpha$ (*Wang et al., 2003*), thereby increasing the suppressive capacity of Tregs (*Wang et al., 2010*). Unfortunately, no significant changes were found in the levels of $CD8^+\alpha 7^+$ T cells and $CD8^+$ Tregs in SCOPD or AECOPD patients. However, Tc10 cells were reduced in both SCOPD and AECOPD patients, revealing a lack of this anti-inflammatory cell throughout the disease course. Consistently, the concentration of IL-10 was reported to be reduced in the sputum of COPD patients (*Takanashi et al., 1999*), and less IL-10 was released from lung tissue of COPD patients after LPS stimulation compared with that of healthy smokers (*Hackett et al., 2008*).

Further comprehensive analyses were performed by calculating the relative proportions of each of the $CD8^+$ T cell subsets in the three groups. In both SCOPD and AECOPD patients, the proportions of pro-inflammatory Tc1 cells were robustly increased, whereas those of Tc2 cells were relatively reduced. Local recruitment of an overwhelming number of Tc1 cells to the lungs and airway mucosa may lead to persistent chronic inflammation; meanwhile, polarization from Tc2 toward Tc1 could promote inflammation through the secretion of IFN-$\gamma$ (*Mosmann, Li & Sad, 1997*). Considering the modestly increased quantity of $CD8^+$ Tregs in SCOPD and AECOPD patients, together with the IL-10-producing capacity of $CD8^+$ Tregs (*Dinesh et al., 2010*; *Suzuki et al., 2008*), we speculate that an increased percentage of $CD8^+$ Tregs may lose their suppressive ability and may be, to some extent, compensating for the significantly low proportion of Tc10 cells in this environment (*Profita et al., 2009*; *Suzuki et al., 2008*). Additionally, given that human $CD4^+$ Tregs are heterogeneous and that an imbalance exists between $CD4^+$ Treg subsets in COPD (*Hou et al., 2013*), we also presume that $CD8^+$ Tregs have similar function and characteristic in COPD patients. In conclusion, these data suggest an insufficiency of anti-inflammatory functions of $CD8^+$ Tregs or Tc10 cells in the context of COPD. However, our analysis only included Tregs and Tc10 cells as anti-inflammatory cells, while other IL-10-producing $CD8^+$ T cell subsets also exist (*Suzuki et al., 2008*). Further studies are required to elucidate the potential autoimmune component of COPD, to determine whether the chronic inflammatory response mainly involves $CD8^+$ T cells and to assess whether this response is similar to delayed-type hypersensitivity such as sarcoidosis.

Some limitations of our study should be acknowledged. Our present study recruits more men volunteers than women donors, which is in line with higher global prevalence of COPD in men than in women. However, recent research has indicated an increased female

susceptibility to smoking-related lung damage, although the findings are controversial (*Sorheim et al., 2010*). The fact that only a few women volunteers are included limits our ability to assess the gender difference in susceptibility in COPD. Moreover, since aging (*Messaoudi et al., 2004*) and GOLD stages may be associated with gradual shifts in CD8+ T cell repertoire, larger sample size is required to ascertain the correlation between CD8+ T cells subsets and the age or certain stage of COPD. Lastly, selection bias of participants cannot be excluded, since different study samples could potentially have affected the results.

## CONCLUSIONS

Our present study indicates the existence of an imbalance of Tc1/Tc2/Tc17 cells in COPD patients, suggesting that COPD is a chronic inflammatory disease characterized predominantly by a Tc1 outburst. Additionally, the imbalance of pro/anti-inflammatory CD8+ T cell subsets observed in COPD patients may be caused by the lack of Tc10 cells and the impaired anti-inflammatory capacity of CD8+ Tregs. However, this report is limited to data from blood samples and lacks information from lung tissue or bronchoalveolar lavage fluids, and much effort is needed to investigate the net changes in both local and systemic compartments.

## ACKNOWLEDGEMENTS

The authors thank Liang Shi for the excellent flow cytometric assistance; Zhi-Jian Ye, Ming-Li Yuan and Wen Yin for their helpful suggestions and discussion; Xia Yang and Dan Yang for their assistance in patient recruitment; and Xiao-Nan Tao for the administrative support.

### Funding

This study was supported by National Natural Science Foundation of China (No. 81370146, No. 81570032 and No. 81500031) and Science and Technology Program of Wuhan, China (No. 2013062301010804). The funders had no role in study design, data collection and analysis, decision to publish, or preparation of the manuscript.

### Grant Disclosures

The following grant information was disclosed by the authors:
National Natural Science Foundation of China: 81370146, 81570032, 81500031.
Science and Technology Program of Wuhan, China: 2013062301010804.

### Competing Interests

The authors declare there are no competing interests.

### Author Contributions

- Long Chen and Gang Chen performed the experiments, analyzed the data, wrote the paper, prepared figures and/or tables, reviewed drafts of the paper.

- Ming-Qiang Zhang performed the experiments, wrote the paper, prepared figures and/or tables, reviewed drafts of the paper.
- Xian-Zhi Xiong conceived and designed the experiments, reviewed drafts of the paper.
- Hong-Ju Liu, Jian-Bao Xin, Jian-Chu Zhang, Jiang-Hua Wu, Zhao-Ji Meng and Sheng-Wen Sun contributed reagents/materials/analysis tools, reviewed drafts of the paper.

## Human Ethics

The following information was supplied relating to ethical approvals (i.e., approving body and any reference numbers):

This study was approved by the Ethics Committee of Tongji Medical School, Huazhong University of Science and Technology (# 2013/S048).

## Data Availability

The raw data has been supplied as Data S1.

## Supplemental Information

Supplemental information for this article can be found online at http://dx.doi.org/10.7717/peerj.2301#supplemental-information.

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
