# Peer review of "Imbalance between subsets of CD8+ peripheral blood T cells in patients with chronic obstructive pulmonary disease"

_PeerJ, doi:10.7717/peerj.2301_

## Round 0.1 · original submission · Major Revisions

While both reviewers find your work of interest, they have raised points that need to be addressed by a major revision. We therefore invite you to revise and resubmit your manuscript, taking into account the points raised.

Reviewer 1 ·

Basic reporting

The authors provide basic description of the CD8+ T cell composition in COPD patients and healthy controls. Using T cell signature surface markers and signature cytokine production following mitogenic stimulation in vitro the authors discriminate canonical T cell subsets in peripheral blood. This contributes to the body of knowledge in COPD research.

The manuscript is generally well written and comprehensible. The figures comprise both exemplary flow cytometric dot plots as well as clear statistical summary plots.

Apparently, the data shown here in the manuscript are derived from the same sample set published earlier (PubMed PMID: 25375131) dealing with the in depth analysis of the CD4+ T cell compartment in COPD patients. This fact should be at least clearly stated in the introduction as this information allows the reader to draw deeper conclusions on the CD8+ T cell data presented in this manuscript.

Experimental design

The studied patient cohort seems well controlled and is age-matched.

The reasoning behind analyzing CD8+alpha7+ T cells should be stated more clearly in the introduction. Especially since there is only a minor population of alpha7+CD8+ T cells and the authors couldn’t detect any significant differences in this cell population between the donor groups. Moreover, the alpha7+ population is disregarded in the summary figure 4. Thus, it only makes sense to show the results for the alpha7 marker if there is a clearly stated expectation for it in context of COPD.

The antibody clones used should be stated in the flow cytometric methods part.

Were dead cells excluded from the analysis by means of a fixable live/dead marker after PMA/ionomycin stimulation? This is important as in vitro T cell stimulation with PMA/ionomycin induces lots of apoptotic cells not producing any cytokines and thereby masking/reducing the percentage of cytokine-positive cells. This should be mentioned in the material and methods section.

Why was CD25 marker used (according to material and methods section) but neither used in the gating strategy nor shown in the data analysis?

Analysis of intracellular cytokines would contribute from showing isotype controls or at least fluorescence-minus-one controls.

Validity of the findings

In figure 4 the authors summarize their flow cytometric analysis by showing the average composition of the peripheral CD8+ T cell pool. However, the figure ignores the presence of naïve CD8+ T cells not being Foxp3+ nor producing any cytokines at all. The presentation of figure 4 would become more conclusive if the true composition of the entire peripheral CD8+ T cell pool would be presented with the CD8+ T cells being Foxp3-IFN-IL10-IL4-IL17- termed as “miscellaneous” or "naive" CD8+ T cells. Of course, it is possible that due to donor variations the total sum may then exceed or undercut 100 %. Therefore, the standard deviations of the averaged subset percentages should be stated in the graph as well.

The data on the CD8+ T cell pool in COPD patients should be discussed in context of the author's findings regarding their data on their previous reports of the CD4+ compartment. This would contribute to the overall interpretation of the true role of the CD8+ compartment in COPD.

Additional comments

Further text related issues:

Lines 160 – 163: This is an over-interpretation

Lines 192 – 193: Sentence should be rephrased to:
In conflict with published findings (Freeman et al. 2015), the percentage of peripheral CD8+ T cells was elevated in AECOPD but not in SCOPD patients.

Lines 193 – 194: Sentence should be rephrased to:
Combined with previous findings that the percentage of sputum CD8+ T lymphocytes was significantly increased at the onset of exacerbations (Makris et al. 2008), we speculate that the increased frequency of CD8+ T cells is limited to the lungs, whereas the expanded CD8+ T lymphocyte population spreads throughout the body during an acute exacerbation.

Lines 199 – 204: Remove semicolon and change beginning of next sentence :
….compared with the stable state (Markis et al. 2008). Another study however reported…

Lines 223 – 226: Sentence should be rephrased to:
Consistently, the concentration of IL-10 was reported to be reduced in the sputum of COPD patients…

Lines 237 – 239: Rephrasing needed:
“…CD8+ Tregs are in a similar situation…” This sounds inaccurate.

Lines: 251 - 253: This is not a valid sentence:
"....research is needed on changes of bith local and systemic. "

Reviewer 2 ·

Basic reporting

No Comments

Experimental design

No Comments

Validity of the findings

No comments

Additional comments

The description concerning the analysis of different CD8+T cells subsets in COPD patients is interesting. However, the authors should address the following questions to improve the manuscript.

1) Authors have described that the demographic details of COPD and NC could be referred to Zhang MQ, 2014. However, I suggest to re-include the table again in this manuscript with an additional detail on the GOLD stages for COPD patients.
2) Although the authors have shown the mean values, the statistical significance by p value is always indicated as p<0.05 throughout the manuscript. Hence, inclusion of respective and specific p values in all the figures, could be more helpful for the readers to understand the depth of significant differences at first glance.
3) Also, I suggest that author should also perform correlation analysis between the subsets of CD8+T cells and age of COPD patients, and the GOLD stages of COPD patients. This analysis will provide with more informative details, if these subsets imbalance is particularly and significantly restricted to the age or certain stage of COPD.
4) The table shows the recruitment of more men COPD patients than women patients and since statistical analysis is not possible to stratify men and women because of just 2 women healthy donor, I suggest to plot an additional graphs by removing all the women from the group to show if the imbalanced effects still exist and significant and is not influenced by minority of women community in the study. Based on the result obtained, authors could discuss little bit more on both introduction and discussion concerning pathogenesis of COPD in men.

---

## Round 0.2 · accepted · Accept

At the level of proof check I kindly ask you to include the following
correction:

line 251 - 253:
"...much effort is needed to investigate the net changes in both local and systemic."
should be rephrased to:
"...much effort is needed to investigate the net changes in both local and systemic compartments.

Reviewer 1 ·

Basic reporting

The authors answered most qustions and remarks placed in the first review process which improved the overall impression of the manuscript.

Experimental design

The reasoning for the alpha7+CD8+ T cell analysis is explained better now. The antibody clones have been stated. The way of live/dead cell discrimination has been described better in the material and methods section and isotype controls for the cytokine staining were added.

Validity of the findings

I asked for the statement of the standard deviations of the percentages of the different T cell subsets in Figure 4 such that in the graphic itself not only the mean value for each T cell subset in the SCOPD, AECOPD and HN group is shown, but also the accoriding standard deviation for each cell subset in a donor groups. These SD values have been actually already calculated by the authors in the "raw data" pdf file. As the percentages given in figure 4 were normalized to sum up to 100% the SD values have to be scaled accordingly. Another way would be to add the according coefficients of variance (CVs) to the figure. This would improve the figure pretty much as it is of importace for the reader to be able to judge the range of varation in the FACS percentages.

Additional comments

Minor corrections:

I still do not understand the rephrased wording in line 251 - 253:
"...much effort is needed to investigate the net changes in both local and systemic."
In my opinion this should be rephrased to:
"...much effort is needed to investigate the net changes in both local and systemic compartments.